# esg2go: A Method to Reduce Bias, Improve Coherence, and Increase Practicality of ESG Rating and Reporting

Isa Cakir [1], Philipp Aerni [1,2,3,*] , Manfred Max Bergman [4,5] and Benjamin Cakir [1]

[1] Center for Corporate Responsibility and Sustainability (CCRS), School of Management Fribourg (HEG-FR), 1700 Fribourg, Switzerland; isa.cakir@ccrs.ch (I.C.); benjamin.cakir@ccrs.ch (B.C.)

[2] School of Management Fribourg, Western University of Applied Sciences (HES-SO), Chemin du Musée 4, 1700 Fribourg, Switzerland

[3] Science and Public Policy Unit, Department of Plant Microbial Biology (IPMB), University of Zurich, 8032 Zurich, Switzerland

[4] Department of Social Sciences, University of Basel, 4501 Basel, Switzerland; max.bergman@unibas.ch

[5] Family Medicine, University of Michigan, Ann Arbor, MI 48109, USA

* Correspondence: philipp.aerni@hefr.ch; Tel.: +41-26-429-64-71

**Abstract:** Rating agencies that assess a company's environmental, social, and corporate governance (ESG) impact have been subject to public and academic scrutiny due to divergent and often biased rating outcomes. Concurrently, an evolving regulatory environment mandates publicly listed companies to report on ESG and climate emissions, taking into account supply chain risks as well. As a result, small and medium-sized enterprises (SMEs) are increasingly asked as suppliers to present a credible sustainability certificate. The esg2go rating and reporting system aims at improving the credibility and practicality of corporate sustainability assessment. It was jointly developed with its users and relevant stakeholders and is based on a calibrated benchmarking system from verifiable data. The rating method enables the measurement and comparison of sector- and firm size-specific sustainability performance. Its underlying adaptive parametrization is derived from a coherent and pragmatic definition of SME sustainability as the 'ability to co-exist'. Our data analyses indicate that our scoring function is able to minimize bias and deliver a fair comparability between SMEs. We conclude that esg2go represents a pragmatic and innovative approach to enhance the fairness and accuracy of corporate sustainability assessment.

**Keywords:** corporate sustainability assessment; rating bias; ESG; SME; SDG; calibrated benchmarking; transparency; coexistence; footprint; handprint; model arbitrage

## 1. Introduction

Global policy initiatives to move toward a more sustainable economy have led to new laws and standards designed to increase the transparency of how businesses manage their environmental, social, and governance risks [1,2]. Consequently, mandatory and standardized ESG disclosure requirements will increase significantly worldwide in the coming years, irrespective of harmonization efforts between different ESG disclosure standards [3].

### 1.1. EU Green Deal Reporting Requirements

The most prominent ESG legislation of 2023 is the EU Directive on Corporate Sustainability Reporting (CSRD) [4], which is designed to make Europe the first climate-neutral continent as part of the EU Green Deal. It requires all publicly listed companies and those with more than 250 employees, a balance sheet total of EUR 20 million, or a turnover greater than EUR 40 million (2/3 criteria) to disclose material sustainability metrics, set targets, report on progress, and provide a third-party audit. The EU directive also applies to EU subsidiaries of non-EU parent companies if they exceed an annual turnover threshold within

the EU. A mandatory audit will be expected, and reporting has to be in line with the unified Sustainability Reporting Standards developed by the European Financial Reporting Advisory Group (EFRAG) [5]. In addition, the CSRD requires companies to disclose information on metrics and targets related to the climate risks of direct emission sources owned or controlled by a company (Scope 1) and indirect emissions resulting from a company's activities, but not from sources controlled or owned by it (Scope 2 and 3).

The implementation of the EU directive will also put indirect pressure on companies that do not meet the 2/3 criteria if they are part of the supply chain of a larger EU-based company that is obliged to report [6,7]. The focus on due diligence in the supply chain will further increase with the proposed Corporate Sustainability Due Diligence Directive (CSDDD) [8]. The CSDDD expects firms to identify, end, or prevent adverse impacts on human rights and the environment in their respective supply chains, buttressed by a multi-layered enforcement structure including civil liability for adverse impacts and directors' responsibilities [8].

*1.2. Harmonizing ESG Rating and Reporting*

ESG information tends to rely on survey-based self-reports of companies or on published corporate sustainability reports. In this context, most companies follow the guidelines of dozens of frameworks and reporting standards created by various international private initiatives, notably the Global Reporting Initiative (GRI), the International Integrated Reporting Council (IIRC), and the Carbon Disclosure Project (CDP). There are also well-known international framework agreements facilitated by international organizations, such as the OECD (Guidelines for Multinational Enterprises, MNEs), ILO (MNE Declaration), the UN Principles Guidelines on Business and Human Rights (UNGP), and the Principles for Responsible Investment (PRI), an investor initiative in partnership with the Finance Initiative of the United Nations Environment Programme (UNEP) and the UN Global Compact.

The most important standard-setter in the United States is the Sustainability Standards Accounting Board (SASB). There are significant differences between the USA and the EU on ESG standards. While the SASB is based on comparability inside one industrial sector, the EU disclosure requirements are on an intersectoral basis. Furthermore, the SASB relies on 'simple materiality' in its assessment of how ESG risks impact a company's performance, while the EU requires a double materiality assessment that also includes the firm's impact on its environment and stakeholders [9].

Global harmonizing efforts achieved a milestone when the IIRC and SASB completed a merger to combine forces under the umbrella of the Value Reporting Foundation (VRF) in June 2021. Subsequently, the VRF and the Climate Disclosure Standards Board (CDSB) reached an agreement with the International Financial Reporting Standards Foundation (IFRSF) to combine these organizations into one global ESG standards-setting body under the IFRSF. Finally, the IFRSF announced the formation of the International Sustainability Standards Board (ISSB) in November 2021 to define a common language for ESG standards [10]. The ISSB seeks to provide a global baseline of ESG-related disclosure standards, consolidating the work of earlier initiatives into a single entity. In June 2023, it issued its standards on general requirements for the disclosure of sustainability-related financial information (IFRS S1) and climate-related disclosures (IFRS S2) [11] following the recommendations of the Task Force on Climate-Related Financial Disclosures (TCFD) [12]. The TCFD recommendations have been endorsed by the G20 and are increasingly implemented by the biggest financial actors. They also seem to have broad support within key international fora such as the Financial Stability Board and the central banks represented in the Network for Greening the Financial System. Based on the TCFD framework, the US Securities and Exchange Commission (SEC) proposed new rules requiring companies to provide detailed information about their handling of climate-related risks and opportunities. The proposed rules will also require companies to measure and disclose greenhouse

gas (GHG) emissions in accordance with the GHG Protocol methodology, the most widely employed international standard for calculating GHG emissions [3].

### 1.3. ESG Ratings under Public and Academic Scrutiny

The global trend toward harmonization and stricter corporate sustainability regulation indicates that ESG disclosure has become imperative, and it will likely reach small and medium-sized companies (SMEs) due to supply chain dynamics [13]. However, the more ESG disclosures become a requisite part of corporate reporting and due diligence, the more they are subject to public and academic scrutiny. In this context, several recent scandals have raised doubts about the accuracy, credibility, and reliability of ESG rating providers [14,15]. Although the EU [16] and the US [17] are currently proposing anti-greenwashing legislation, it is unlikely that the challenges in measuring and comparing ESG performance can be addressed effectively in view of research unveiling a general lack of consistency and coherence of sustainability rating and reporting systems [18,19].

#### 1.3.1. The Problem of ESG Divergent Rating Outcomes

According to recent empirical studies [19–22], there is a growing divergence of outcomes among established ESG rating and reporting systems, which has led to confusion and a general distrust in the ESG rating industry. The divergence between ESG ratings can be mainly explained by differences in measurement (i.e., the same object is measured in different ways) and aggregation (different rules of aggregation) as well as a lack of commonality in the definition of environmental, social, and governance components [23]. This also explains the large gap in accuracy between sustainability rating and credit risk rating. Berg et al. [19] showed that ESG ratings from five prominent data providers correlated between 0.38 and 0.71, whereas the correlation for credit ratings between the largest credit risk agencies is 0.99. The relatively low correlation of ESG ratings is also related to the fact that the claimed outcomes cannot be subject to a basic accuracy test. Such an accuracy test exists in credit risk evaluation: it is the likelihood of default, leading the credit rating scale to flow naturally from safest to defaulted for all businesses [24].

Rating divergence leads to an arbitrage in the broad sense that companies tend to select the rating that makes them look best. This increases the risk of greenwashing by concealing problematic aspects of a business. Since most rating tools are proprietary, it is also impossible to scrutinize the methodology underpinning the ratings. Most ESG ratings are at least partially driven by commercial interests and are thus methodologically non-transparent. Rating agencies market diverse products and services such as sustainability indices, sector and thematic research reports, benchmarks, etc., which may increase their bargaining power but also put into question their ability to provide unbiased concepts of sustainability [25].

#### 1.3.2. Different Mindsets behind ESG Footprint Assessments and SDGs

All established ESG rating and reporting systems primarily assess the risk of business activities for society and the environment (Footprint) and how to reduce it. In contrast, the Sustainable Development Goals (SDGs) and Agenda 2030 [26], which were approved by United Nations General Assembly in 2015, also acknowledge that business may become part of the solution if companies are provided with incentives to invest in sustainable innovation [27–29]. Whereas corporate sustainability risk assessment is associated with the footprint of a company, the ability of a company to produce positive external effects for society and the environment through its long-term investments is associated with its corporate handprint [30,31]. A company's handprint refers to core business activities that also generate positive externalities for society and the environment, which are usually not captured in the footprint assessment. The positive external effects for society and the environment generated, for example, through an innovation that substitutes an existing unsustainable product or process with a more sustainable one, may also have to be taken

into account [31]. In other words, an assessed negative corporate footprint may have to be balanced against a potential positive handprint [32,33].

The lack of acknowledgment of the contribution of their core business to sustainable and inclusive change may be one of the reasons why the majority of SMEs have refrained so far from embracing a particular sustainability rating and reporting system [34].

### 1.3.3. Rating Biases Due to Data Quality Problems and Incoherence in the Definition of Corporate Sustainability

Research has revealed that ESG ratings suffer from biases that remain largely unaddressed, such as the firm size bias [35–37]. In contemporary ESG ratings, smaller firms consistently perform poorer than larger firms across different ESG scores. In addition, a data quantity bias has been identified, showing that a company's available resources for providing ESG data and the availability of a company's ESG data tend to largely determine a company's assessed sustainability performance [38,39]. In other words, the extent to which and manner in which data is provided matters. With regard to the latter, a company that can afford an experienced consultant is likely to get a better rating than one that cannot. This implies that ESG data may not be altogether reliable because of deficiencies in quantity, consistency, and quality.

The quantity bias may also be related to the industry category or sector bias. For example, the large credit information provider CRIF claims that it was able to provide 470,000 Swiss SMEs with ESG certificates based on estimations that rely on publicly available data as well as ESG data provided by the firms, if available [40]. Since no ESG data is available from most non-listed SMEs, its ESG evaluations revealed, quite predictably, that most companies involved in service activities—low in emission intensity and environmental impact—are sustainable while those involved in mining, agriculture, and other resource-intensive industries are not. This sector bias has also been confirmed by other studies [39,41].

Apart from the identified entrenched rating biases, the great variation in definitions of corporate sustainability that underpin the different rating approaches also undermine the credibility of corporate sustainability assessments [42,43]. In this context, studies point at inconsistent terminology [44], unstandardized and subjective ESG scores [45], and materiality problems [46] that make it impossible to subject ESG metrics to rigorous empirical testing. This is one important reason why the impact of corporate sustainability on firm performance remains inconclusive [47].

### 1.3.4. Addressing SME Constraints and ESG Rating Biases through esg2go

Despite these challenges, stakeholder theorists argue that corporate sustainability activities may eventually pay off because they improve reputation as well as freedom to operate, which could attract more resources, investment, and skilled employees [48,49]. Many SMEs share this view in anticipation of the expected growing regulatory pressure that may affect them indirectly as suppliers of large companies that must comply with the CSRD, ISSB, and/or TCFD, to name only the few discussed above [50–52].

However, there are also disincentives to employ proprietary sustainability rating tools: apart from their inherent biases discussed earlier, they are also costly and time-consuming [53].

In spring 2023, the EU proposed a new directive designed to avoid the misuse of green claims [16]. It is meant to tackle numerous biases and conflicts of interest in the ESG rating and reporting industry [54]. However, it will be difficult to implement this directive without an affordable ESG rating tool that enables a more science-based, practical, and transparent approach to measuring and comparing sustainability performance across industry categories and firm sizes.

We first discuss the purpose and mindset of esg2go in Section 2. In Section 3, we present a coherent definition of corporate sustainability and the rating approach from which it is derived. Section 4 presents the methodology behind the implementation of

the rating approach, including the contextualization of the KPIs, the scoring function, the adaptive parametrization process, and the calibration of benchmarks. Section 5 covers the empirical validation of the ability of the scoring function to reduce variation among SMEs through the formation of sustainability classes. Section 6 discusses the findings in the context of prior research on biased outcomes of ESG ratings, and Section 7 concludes by highlighting the main advantages of esg2go and discussing its prospect to become an internationally recognized corporate sustainability rating.

## 2. Purpose and Mindset behind esg2go

'esg2go' is a sustainability rating and reporting system that aims at reducing bias while improving coherence and practicality in corporate sustainability assessment. It does so through a rigorous rating methodology that enables the measurement and comparison of sustainability performance, taking into account firm size, industry category, and win–win potential for the firm, as well as for sustainability.

However, esg2go is only addressing sustainability on the firm level. As such, it cannot be compared to product- and process-specific sustainability evaluations. It is also not meant to replace theme-specific (focus on specific environmental and social aspects) or industry-specific sustainability standards.

### 2.1. Addressing the Data Quality Problem

The esg2go rating tool relies on the data entered by user firms, which complete an online questionnaire covering the dimensions E, S, and G, as well as an optional handprint assessment. Consequently, it is the user firm that is accountable for the veracity of the input data. The input consists of concrete and verifiable data that are found either in the accounting system or can be obtained with respective providers (energy, water, waste disposal), as well as discrete answers ('yes', 'no', 'in process') referring to the existence of specific corporate documents or policies (see questionnaire with the list of indicators in the Supplementary Material). Prior to data entry, the user firms sign a data protection agreement with the esg2go rating provider, who is committed to strict data confidentiality.

### 2.2. Ensuring Firm Ownership While Enabling Data Quality

Since the user firm remains the official owner of the data, it ultimately decides who has the right to see it and in which degree of granularity. In return, as an academic research institute, the esg2go rating provider (CCRS) has the right to use the data for research purposes as long as the user-identity remains strictly anonymous. This right enables the CCRS to conduct sustainability research in the real economy based on reliable data. Furthermore, it will also help to continuously improve the accuracy of the rating over time.

### 2.3. Sustainability Understood as a Process Rather Than a State or a Product

esg2go provides the user firm with a first base assessment of the actual sustainability performance. As such, the fully automated esg2go rating report offers SMEs a mirror in the form of a spider graph that shows where they currently stand in regard to 10 scored key areas compared to their peers (benchmark). Based on this knowledge about their actual sustainability performance, the semi-automated esg2go reporting system provides them with the opportunity to explain their rating outcome and define measures and set targets for relevant key areas. This makes it possible to track ESG performance in a fair and consistent way and to monitor the possible gap between actual and target states over time. In this context, corporate sustainability is defined as a process rather than a state, represented through a traffic light procedure (red, orange, green). Understanding sustainability as a process provides a differentiated view and regards concrete improvements over time as being more relevant than an aggregated judgment of the current performance.

esg2go also addresses the concern that the overall ESG score may conceal large disparities between 'E' and 'S' performance [55] by clearly separating scores obtained in key

areas in the dimension 'E', 'S', and 'G'. Disparities between 'E' and 'S' performance are thus revealed and can be addressed as part of a firm's sustainability transformation.

### 2.4. A Qualitative Handprint Assessment as Optional Input

A comprehensive sustainability rating should not just assess the potential sustainability risks of business but must also take into account that business can be part of the solution. Contemporary handprint assessments that capture such positive external effects are primarily focused on product assessment [56]. esg2go offers a qualitative handprint assessment on the firm level based on a set of questions regarding the potential positive side effects of the core business activity on society and the environment ('yes' or 'no' answers). If a question is answered with 'yes', then the corresponding documentation has to be uploaded and will be reviewed by the established independent esg2go expert committee. The committee then decides if a certain improvement of the overall esg2go score is justified. The handprint ensures that the esg2go risk assessment is also in line with the SDG spirit, with its emphasis on sustainability as a business opportunity [29].

### 2.5. The Mindset of esg2go

The mindset underpinning the esg2go framework is pragmatic in the sense that it is not assumed that SMEs aim to become sustainability champions for its own sake. Instead, there must be an expected 'return on investment' which internally justifies the mobilization of company resources. Resource mobilization must be based on a 'win–win' mindset, where measures taken to improve the overall corporate sustainability performance are not just good for society and the environment, but also for the company. After all, a company that invests available resources in expensive sustainability measures may ultimately not be sustainable if it leads to a general neglect of its core business activities and eventual bankruptcy. Therefore, the 'governance' part of esg2go also takes into account the financial condition of a company and thus goes beyond non-financial reporting.

#### Defining Corporate Sustainability as the Ability of a Firm to 'Coexist'

A credible sustainability rating that captures and monitors the sustainability performance of SMEs in different contexts over time must start with a definition of sustainability to which all parties can agree, including the SMEs that are expected to use the tool. Otherwise, the gap between practice and research in sustainability assessment will remain [57].

In a report published in 1987 entitled 'Our Common Future' [58], the World Commission on Environment and Development provided a well-established general definition of sustainable development related to intergenerational equity, which was approved by the UN General Assembly. The 17 UN SDGs build on this concept, but also include the need for 'inclusive growth', emphasizing the importance of sustainable technological and economic change through institutional framework conditions that enable 'business to become part of the solution' [59]. The need for inclusive and sustainable economic change represents the relevant link to corporate sustainability. Firms are the primary engines of job creation, innovation, and income generation and therefore play a potentially crucial role in economic empowerment, poverty reduction, and sustainable and inclusive change [29]. At the same, the SDGs also recognize that business activities may generate risks for society and the environment that have to be addressed by regulation and self-regulation.

The practical definition of corporate sustainability used for the esg2go framework builds on the assumption that the investment in corporate responsibility must pay off in the long run. The concept of materiality is indirectly based on the idea that there must be a return on investment in the improvement of corporate sustainability, and some methodologies have been developed for certain industries to assess it [60,61]. However, materiality has not yet been linked to a pragmatic definition of corporate sustainability that is required to narrow the divergence of rating outcomes and track sustainability improvements over time. For SMEs, investments in sustainability should be supported through a favorable

institutional environment and be recognized by the relevant stakeholders in business and society [62].

In this context, most SMEs pursue a pragmatic approach to corporate sustainability, with a primary focus on responding to changing expectations articulated by the relevant stakeholders (e.g., employees, customers, investors, authorities) on which the business depends. If stakeholder expectations for ESG disclosure rise in response to stricter regulatory requirements, SMEs will re-assess the return on a sustainability assessment [63].

## 3. A Coherent Definition of Sustainability and How to Capture it in an Extended Balance Sheet

Corporate sustainability is ultimately about 'the ability of a firm to coexist', or its ability to respond to the changing expectations of the stakeholders on which the firm's business ultimately depends. It is associated with a systematic business approach and strategy that takes into consideration the long-term social and environmental impact of all economically motivated drivers of a firm minding wider societal concerns [64].

By embracing the definition of corporate sustainability as the ability to coexist, the esg2go sustainability assessment for SMEs focuses on the win–win potential of sustainability. This win–win potential for SMEs may decrease with growing expenses required to comply with due diligence as well as ESG and climate disclosure regulatory requirements [65,66]. SMEs have limited resources at their disposal and are thus confronted with 'trade-offs'. They cannot excel in all areas of sustainability but must set priorities.

### 3.1. Incorporating a Sustainability Dimension into Credit Risk Rating Methodology

esg2go makes use of the know-how in actuarial sciences and general credit risk evaluation, which takes into account long-term survival drivers beyond contractually defined liabilities. These may be regarded as additional liabilities related to the impact of a company on society and the environment. As such, this approach captures the co-viability of a company or probability of coexisting in the future. In this context, sustainability risk assessment is primarily a more complex form of risk management that captures the long-term risk drivers related to societal expectations. This also requires close collaboration with experts from other fields.

### 3.2. Adding System Complexity and an Ethics Dimension to the Balance Sheet

The selection of KPIs and the corresponding benchmarking process are based on improvements in corporate sustainability, measured in a systematic and science-based way, which also enable a company to move toward a higher order of complexity. A higher order of complexity takes place if the system in which a company operates has energy, stabilizing factors, and capacity. This mindset is discussed by Hidalgo in *Why Information Grows* [67]. While 'energy' is about creativity and the generation of ideas, 'stabilizing factors' refer to the filters and structures selecting or discarding ideas, including the creative products derived from them, in a systematic way. 'Capacity' refers to the extent to which a system is able to move toward a higher degree of complexity in consideration of available capacities and resources. However, without taking into account 'ethics' as an additional component in the balance sheet (the fair way of taking and giving, also within generations), the strengths induced by energy, stability, and capacity may also result in harmful societal outcomes.

These considerations give rise to additional balance sheet items induced by consensus and rules associated with ethics in society, which ultimately determine a firm's long-term license to operate and, with it, its ability to co-exist. This mindset is illustrated in Table 1.

**Table 1.** Aspects of sustainability and extended balance sheet items.

| *Aspect 1: Combining Competitiveness With Sustainability* | | | | |
|---|---|---|---|---|
| Dynamics of information growth: it is rather egoistic view that does not consider | | | *Interaction with other systems or players* | |
| Energy | Stability | Capacity | *Ethics* | |

| *Aspect 2: Stakeholder Impact View* | | | | | | | |
|---|---|---|---|---|---|---|---|
| Priority 1: Liabilities—solvency view | | | | | Priority 2 | **Sustainability-driven extended balance sheet items** | |
| Clients | Employee | Creditors | Tax authorities | Other contractual liabilities | Shareholders | **Society** | **Environment** | **Future generations** |

Orange represents self-regarding concerns, Green represents other-regarding concerns.

The additional 'balance sheet' items in Table 1 supplement the short-term survival in a competitive business environment as assessed in credit risk rating with a co-survival dimension, giving rise to stakeholder impacts that are mostly concerned with long-term survival drivers. It assesses the sustainability of SMEs by their contribution to the expectations expressed by an additional set of relevant stakeholders, which can be mapped to sustainability targets. For parametrization purposes in the benchmarking process, it is important to select positions that can be assessed with an adequate level of granularity. Table 2 displays such positions without claiming completeness.

**Table 2.** Example of extended balance sheet items.

| **Coexistence Balance Sheet Items** |
|---|
| Harmful emissions and climate |
| Waste |
| Real economy (unemployment, GDP, etc.) |
| Discrimination on all levels |
| Youth education and training |
| Social security |
| Reintegration of people with handicaps |
| Protecting health and safety |
| Protecting human rights, including economic rights |
| Etc. |

## 4. Methodology

In this chapter, we first discuss the evolution of the esg2go methodology from a first prototype to a user-friendly rating product that is based on a set of indicators in the dimensions 'E', 'S', and 'G' that have been identified as relevant, measurable, and practical in a open and collaborative process with diverse stakeholders. Subsequently, we outline how model arbitrage and bias are minimized through numeric and contextualized KPIs, illustrated by means of a concrete example. Finally, we show how the scoring of KPI enables measurability and comparability of sustainability performance across industry categories and firm sizes through a calibrated benchmarking process.

### 4.1. Origination of the esg2go Methodology

The esg2go framework has been under development since 2018. It started with the development of a prototype tool supported by the Renaissance Foundation, a private equity firm that invests on behalf of Swiss pension funds in Swiss SMEs. The overall aim was to produce an effective rating system for corporate sustainability performance that would be useful to SMEs in Switzerland.

### 4.1.1. Transdisciplinary Selection of ESG Indicators

In collaboration with numerous experts in various fields of sustainability and the organization Swisscleantech [68] as a content development partner, ESG indicators were selected from a universe of indicators in the environmental, social, and governance dimensions. The selection of indicators was based on the following criteria: the indicator must be based on verifiable facts (measurability), significant (relevance), and easy to use (practicality). Redundancy would have to be avoided. The selected indicators in the areas of E (environment), S (social), and G (governance) were then submitted to experts in the respective fields for feedback. Subsequently, they were tested in an iterative process with SMEs. Whereas the indicators chosen to capture the environmental (E) and social (S) dimension of corporate sustainability form the core of the rating, governance indicators (G) are primarily used to contextualize E and S indicators.

Regular workshops were held in which relevant stakeholders as well as independent experts were invited to provide feedback on the prototype version and the selection of indicators. This open and collaborative process was essential for setting a broadly accepted standard in the real economy that takes into account the fact that SMEs tend to face more resource constraints and therefore prefer a pragmatic approach to sustainability management.

### 4.1.2. Converting a Prototype Tool into a User-Friendly ESG Rating Tool

The methodology was converted into a user-friendly online ESG rating tool in November 2021 to offer a first test version of the jointly designed esg2go online platform (www.esg2go.org (accessed on 7 December 2023)). Within a time span of 5 months (November 2021 to March 2022) and due in part to a national media coverage, 250 companies registered and about 120 entered company data. These input data observations were used in combination with officially available ESG-relevant data by industry category to develop a calibrated benchmarking system. It produced relatively robust estimations for the first time when launched as a minimal viable product (MVP) in June 2023.

### 4.2. *The Basic Rationale behind the Methodology*

With the launch of the MVP, esg2go became the first corporate sustainability rating tool that measures and compares corporate sustainability across industry categories and firm sizes based on an adaptive calibrated benchmarking system. It displays a company's position compared to the benchmark of the respective key area (KA) based on the distance to an ideal target value. The scores of the 10 KAs in the dimensions E, S, and G are presented in the esg2go output in the form of a spider graph for each dimension, highlighting where the performance is above or below its respective benchmark.

The holistic score of all Swiss companies with a minimal age of two years and at least two employees can be determined based on the input data of the companies. Companies with more than 250 employees may also use esg2go. However, their main concern is ESG disclosure due to new regulatory requirements (e.g., CSRD in Europe). esg2go can assist in meeting these new requirements through the creation of so-called filters, which enable a company to transcode the results of the esg2go rating report as text into the corresponding answer boxes of the questionnaires of the established reporting standard, such as the GRI or Deutscher Nachhaltigkeitskodex (DNK).

In addition, esg2go responds to the growing demand for greenhouse gas emission reporting. Based on data entered in the key area 'energy and waste', esg2go discloses direct and indirect $CO_2$ emissions in accordance with the Greenhouse Gas Protocol (GHP) [69], which requires emissions to be reported in three 'scopes'. Scope 1 emissions are direct emissions from owned or controlled sources. Scope 2 emissions are indirect emissions from the generation of purchased energy. Scope 3 emissions are all indirect emissions (not included in Scope 2) that occur in the value chain of the reporting company, including upstream and downstream emissions. Scope 3 (first tier supplier) reporting requires few additional questions in esg2go to be answered. The degree of detail provided in the answers determines the level of granularity (disclosed as low, average, high). The resulting climate report fully

complies with the SBTi reporting requirements for SMEs (https://sciencebasedtargets.org/ (accessed on 7 December 2023)).

### 4.2.1. Ruling out Model Risk and Avoiding Model Arbitrage

Overall, esg2go aims at ruling out model risk to ensure that it is not based on inaccurate or even false claims. In this context, model risk is associated with model arbitrage and the risk of false predictions related to the lack of available data required to achieve the target of high granularity in the model.

The model framework continuously rules out model arbitrage by effectively capturing and rating corporate sustainability performance. It does so by increasing the granularity of its parametrization, which enhances discrimination power and ensures fair comparability. A first contextualization of input takes place through the design of key performance indicators (KPIs) that are based on data provided by participating SMEs. They form the foundation of the parametrization process.

Model arbitrage can take place by outsourcing certain business activities that are emission-intensive or intensively using natural resources. In the course of continuous parameter updates, esg2go takes direct outsourcing items, such as cloud solutions, district heating, and all-inclusive rent, into account to address this type of model arbitrage.

The risk of false predictions is addressed through the adaptive nature of the esg2go rating framework. It is designed to improve in accuracy over time with the growing availability of firm data. In other words, the error tolerance interval around the benchmark continuously narrows as the number of firms using esg2go increases. In this sense, the esg2go framework is understood as a balanced learning tool that improves as more firms participate over time.

### 4.2.2. Bonus Questions to Capture the 'Handprint'

The rating tool offers optional bonus questions, which capture the so-called 'handprint' of a company (see Section 1.3.2). Response options to questions in the environmental and social dimension of sustainability are binary ('yes' or 'no'). Each item on the list that has been marked 'yes' has to be documented as explained in Section 2.4.

### *4.3. The Calculation and Contextualization of KPIs*

esg2go is built on 60 KPIs to enable a robust and fair sustainability metric based on 10 key areas (KAs) in the three dimensions E, S, and G. The KPIs are functionally classified as numeric KPIs and status KPIs. Numeric KPIs use inputs that require numerical values. They are the result of the contextualization of inputs and are expressed relative to pre-set references. Status KPIs are based on inputs that indicate the existence of certain documents in the company. The status options are 'yes', 'in process', and 'no'.

Each contextualization is based on valuation criteria such as voluntariness, model arbitrage possibilities, comparability aspects, and other boundary conditions so that their scoring reflects a fair picture of SMEs with respect to KPIs.

An example of the contextualization of a KPI which also illustrates the notion 'model arbitrage' is the social KPI CML (contribution for maternity leave) defined as follows.

$$\text{CML} = \frac{\text{NF}}{\text{NE}}(\text{M} - \text{OLM})$$

where:

NE = the number of employees
NF = the number of female employees aged between 16 and 45
M = the minimal length of the company's maternity leave in weeks
OLM = the official length of maternity leave in weeks (=14 weeks in Switzerland).

The input number of female employees aged between 16 and 45 includes all female employees who are between the ages of 16 and 45 (NF). The input minimal length of the

company's maternity leave refers to the company's minimal number of weeks for maternity leave given to female employees by the company (M). During maternity leave, female employees continue to be paid while they stay at the hospital or at home to care for and bond with their children. The official length of maternity leave mandated by the Swiss government is 14 weeks (OLM). In this KPI, ALM is compared with the number of weeks the company is providing to their female employees (M).

The KPI considers only female employees aged between 16 and 45, since that is of child-bearing age. Furthermore, the KPI reduces the risk of model arbitrage so that companies with zero or very few female employees are not unduly rewarded by means of our scoring as 'generous' with respect to maternity leave.

Consequently, the KPI assesses how generous the company is in providing their female employees with additional weeks of maternity leave beyond what is required by law. Maternity leave ensures that working women do not have to choose between a career and having a family and can maintain financial stability more easily during pregnancy and the weeks following childbirth. This may also have an indirect positive impact on other KPIs. It increases, for example, the likelihood of a higher and more stable degree of gender diversity within the workforce on every level of the company structure. This may again have a positive impact on the productivity of the company. Furthermore, employer loyalty and employee morale are also positively affected when female employees can enjoy additional weeks of maternity leave.

In this context, the KPI CML illustrates the principled approach leading to numeric KPIs. Other examples of numeric contextualized KPIs are $CO_2$ emissions, holistic diversity, and employees managed by objectives in the main areas E, S, and G, respectively.

### 4.4. Scoring Function of KPIs

The scoring of KPIs is a crucial module of the esg2go framework. Its primary purpose is to minimize variation resulting from the heterogeneity of SMEs and their business activities so that a level of homogeneity can be achieved, on which basis the measurability and comparability of sustainability performance becomes possible.

Given a fixed numeric KPI, it should order SMEs according to their positions within an interval defined by two extremes—'target' and 'NoGo'—which represent the best and worst possible states, respectively. The score should also indicate whether an SME has a satisfactory KPI value, which we indicate with the notion of the benchmark representing the market practice of the sustainability class that has been assigned to the user firm. A transparent approach for this purpose is the piecewise linear function displayed in Figure 1. The graph shows the score on the *Y*-axis and the respective KPI value on the *X*-axis. The scoring function assigns an observed numerical KPI of the respective SME a score between 0 and 100 and takes the benchmark value of 60 when located in the error tolerance interval. Global benchmarks that do not take into account the context in which SMEs operate tend to be 'unfair' because they produce biased outcomes. After all, deviations in the numerical KPIs are driven by the nature of their business. Calibration by sustainability classes takes into account firm sizes and service categories. In this context, we rely on NOGA codes, which are the official industry categories in Switzerland, and other publicly available statistics. Even though they are suitable for a major part of the numeric KPIs, the very nature of the business may require additional information whenever corresponding data is available, as well as a more granular classification approach for some KPIs. However, due to the inperfections of available data and other model risks, the first approximation of the benchmarks may not address all uncertainties regarding the nature of the business. The error tolerance intervals take this into account. They capture model risks by taking into account the variance observed in a numerical KPI: the higher its variance within their sustainability class, the larger the error tolerance interval. The variation is expected to decrease eventually as more data become available.

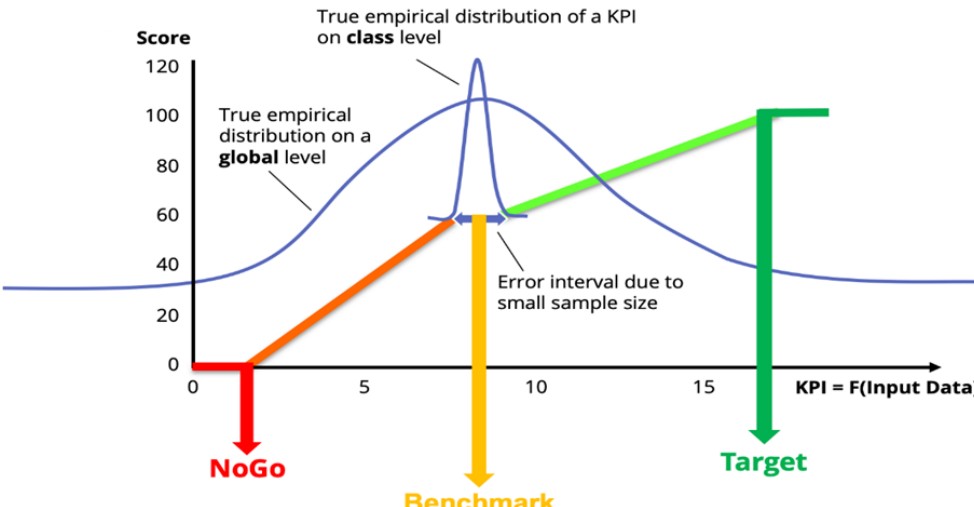

**Figure 1.** Scoring function.

The parameters of the benchmark, target, and NoGo therefore depend on a firm's respective sustainability classes (defined by service category and firm size) with respect to specific KPIs. Benchmarks are determined in a top-down approach through observations of publicly available data and deduction processes wherever required.

The scoring function of a numerical KPI $⟦SF⟧$ is based on five parameters which we will denote as NG (NoGo), BM (benchmark), T (target), EL (left error tolerance), and ER (right error tolerance). These parameters are designed to achieve a fair comparability in consideration of the sustainability class of a SME.

The status KPIs have three non-numerical values: 'in place', 'in progress', and 'no'. The scoring function (SF) of such a KPI has no parameter and is straightforward. It takes values 100, 60, and 0 for 'in place', 'in progress', and 'no', respectively.

The score of a KPI (numeric or status) can hence be defined by the following general formula:

$$\text{Score(KPI)} = \begin{cases} \text{SF}_{\text{NG,BM,TG,EL,ER}}(\text{KPI}) \text{ if KPI is numericl} \\ \text{SF(KPI) If KPI is status} \end{cases}$$

With respect to a numeric KPI (as portrayed in Figure 1), companies performing on a level considered industry practice are scored with 60. The 'higher' and the 'lower' performers are scored between 60–100 and 0–60, respectively.

*4.5. Parametrization*

The two main steps of the parametrization process are scoring parameters followed by aggregation weights as illustrated in Figure 2. The scoring parameters aim at rendering KPIs comparable in a balanced way through the calibration of benchmarks. This is based on two steps; a top-down approach and a sustainability class-dependent benchmark based on an amalgamation procedure mixing the top-down parameters with own observations. Sustainability classes primarily reflect a company's firm size and service category. Once the scoring parameters are set, aggregation weights are determined in three stages using three main criteria: relevance, prioritization, and technical normalization process. The determination of the aggregation weights allows for a holistic view on different reporting levels, also taking into account the win–win potential of a KPI. Figure 2 displays the roles of these two parameter components in the valuation process:

**Figure 2.** Roles of scores and weights in the valuation process (general case).

As for classifications and benchmarking purposes in the process of parametrization, companies are attributed to one of the 18 'sustainability classes' that were formed based on criteria such as the firm size (six size categories) and service categories (three service categories).

Initially, SMEs are split in three size categories: micro, small, and medium (up to 250 employees in accordance with the Swiss definition of an SME). The three categories depend mainly on the top-down accident ratio of the NOGA-based industry category [70]. Later in the process, these three size categories are extended to six categories by splitting the micro companies (companies with less than 10 employees) into two categories creating two additional size categories for companies that exceed 250 employees.

There are cases when we use information beyond these general criteria to assign a company to a particular sustainability class. For example, 'employer loyalty' (KPI in S) takes into account the company's age because older companies are more likely to have higher average work years of their employees than young companies.

Some KPIs may require high granularity regarding the NOGA industry category level to enable the assignment of a company to a particular sustainability class. However, if the number of observations in a NOGA industry category is low, a statistically justified mixture may not be possible. In this case, our estimations are conducted in a purely top-down manner for each NOGA code making use of the publicly available data. For example, data on emission intensity by NOGA industry category is available. This can then be used for the KPI 'CO$_2$ emissions due to energy consumption' (a KPI in main area E).

4.5.1. Calibration of Benchmarks

The calibration of benchmarks, which is part of the scoring parameters, is based on two steps; a top-down approach and a sustainability class-dependent benchmark based on an amalgamation procedure mixing the top-down parameters with our own observations.

For each KPI, the benchmarks are the most crucial parameters of the scoring function and should reflect the 'best estimate' of market practice (the estimation which makes use of all the available information in the most unbiased way) taking into account industry practice within the respective sustainability class, whereas targets and NoGos are determined by law or by the consensus in society (e.g., climate goals of Switzerland), backed up by expert-based knowledge.

*Top-Down Approach*: A top-down estimation to calibrate the benchmarks of the scoring function starts with the study of the secondary literature related to each KPI. Since it is often the case that the secondary literature does not offer direct information exactly synchronized with the contextualization of the KPIs, we must define workarounds to address this problem. This requires different sources and a chain of logical steps; in this context, the gathered data enters a pipeline whose endpoint is the benchmark reflecting the contextualization driving each KPI. For some KPIs, the granularity can be kept at a global level. These are referred to as 'global benchmarks'.

As regards 'sustainability class'-dependent benchmarks, these are calibrated from their top-down counterparts by means of amalgamation procedures.

*Sustainability Class Dependent Benchmark*: As previously mentioned, the benchmarking process of some KPIs is treated globally whenever these have sufficient statistical discriminating power themselves, or, by their very nature, do not require granularity, such as the KPI 'employee dynamic index', which measures the contribution of SMEs to employment. The rest of the KPIs can be treated in a standard manner by amalgamating our top-down benchmarking process with the observed SME data.

*Amalgamation of SME Data with Top-Down Benchmarks*: For each of these KPIs, we start from the related top-down benchmark (TD) and mix it cautiously with the observed averages of each KPI in each sustainability class (SC) through a convex linear combination:

$$\text{BM}_{\text{SC}} = \alpha_{\text{SC}} \overline{\text{KPI}}_{\text{SC}} + (1 - \alpha_{\text{SC}}) \text{TD}$$

Here, $\alpha_{\text{SC}}$ depends on the discriminating power of the classification of SMEs, which is reflected by statistical key figures (such as variance between the classes and variance within the classes) and finalized by an additional probabilistic layer to ensure robustness. This procedure yields one benchmark per class for each numeric KPI.

### 4.5.2. Aggregation Weights

Once scoring parameters are defined, the aggregation module scores weights on a different level. The weights of KPIs are scored in three stages using three main criteria: relevance, prioritization (the degree of incentivization of a win–win scenario), and a technical normalization process.

*Weighing by relevance* starts with a size-dependent filter, followed by an assessment of the net impact on an environmental and/or social balance sheet position, as discussed in Section 3, as well the impact on a firm's business.

*Weighing by prioritization* takes place once a KPI passes the relevance filter. It assesses to what extent the net impact on society and the environment also generates added value to the core business (win–win). If a more favorable net impact comes at the expense of the business of the SME, priority for the respective KPI is lowered.

*The technical normalization process* normalizes the priorities to obtain the final weights.

## 5. Addressing Rating Biases: Empirical Validation of esg2go Sustainability Classes

The goal of this section is to prove that our holistic approach controls for class-dependent bias by showing that the empirical variance of the holistic esg2go score between the classes of our sample is negligible compared to the total variance of our sample. It confirms that the formation of our 18 sustainability classes and the parametrization thereof remove class-dependent drivers or bias in the holistic scores.

*Sample Size*: As of July 2023, there are approximately 500 SMEs registered on esg2go, but only approximately 250 entered data. A total of 103 companies completed all parts and obtained a score in all 10 key areas (KAs). These observations were then assigned to the 18 sustainability classes. For now, there are only seven classes with more than one observation, three classes with one observation, and eight classes did not contain any information yet. As a result, 100 observations can be used to conduct the test, which examines if the limited sample allows one to infer the unbiased nature of the sustainability classes and their associated benchmarks.

*Hypothesis testing*: We test the hypothesis that esg2go generates no bias with respect to its sustainability classes. To support this claim, we rely on the class variance ratio (CVR), our monitoring statistical indicator, which reflects the share of variance absorbed by the variances within the classes:

$$\text{CVR} = \frac{\text{E}[\text{Var}[\text{S(SME)}\,|\,\text{C}]]}{\text{E}[\text{Var}[\text{S(SME)}\,|\,\text{C}]] + \text{Var}[\text{E}[\text{S(SME)}\,|\,\text{C}]]} = \frac{\text{E}[\text{Var}[\text{S(SME)}\,|\,\text{C}]]}{\text{Var}(\text{S(SME)})}$$

In this equation, S(SME) represents the holistic score of an SME. The variance equation suggests that the total variance of a randomly selected score of an SME is expected variance

absorbed by the classes (denoted by E[Var[S(SME)|C)]]), plus variance of expected scores given the class of the SME (represented by Var[E[S(SME)|C)]]). Hence, CVR reflects the share of expected variance within the classes. This implies that the higher the CVR, the lower the variance between the classes and, hence, the lower the likelihood of bias driven by the classes. Intuitively, a perfect unbiased scoring with respect to a classification removes all systematic class-specific drivers so that only the individual performance of the SME plays a role in scoring. In this ideal case, CVR tends to one as the size of the observations tends to infinity.

The observed CVR amounts to a very high 94.7%, but is this observed CVR satisfactory enough to justify our stated hypothesis—namely, that esg2go generates no bias with respect to its sustainability classes? We will answer this question with the aid of Monte-Carlo simulations. We first generate a large number of portfolios of unbiasedly scored SMEs with exactly the same number of SMEs in each class as our observed sample (n = 100) by means of identical, beta-distributed, and independent random variables. In more than 99% of the cases, the simulated CVRs are lower than 93%. This shows how rigorous the CRV threshold of 93% is. It is lower than our actually observed class variance ratio of 94.7%. Hence, we can clearly reject the claim that esg2go allows for more than a marginal bias. We conclude that, given current information and data, our classification, and all class-dependent parameters (score parameters and their respective weights), deliver fair comparability between SMEs.

For the sake of completeness, we introduce a controlled increasing random bias and expect an increase in the probability of getting a lower CVR than 93%, as presented in Figure 3. The *X*-axis in the diagram represents expected value (C) of the controlled random marginal bias operating on the alpha parameter of a beta distribution.

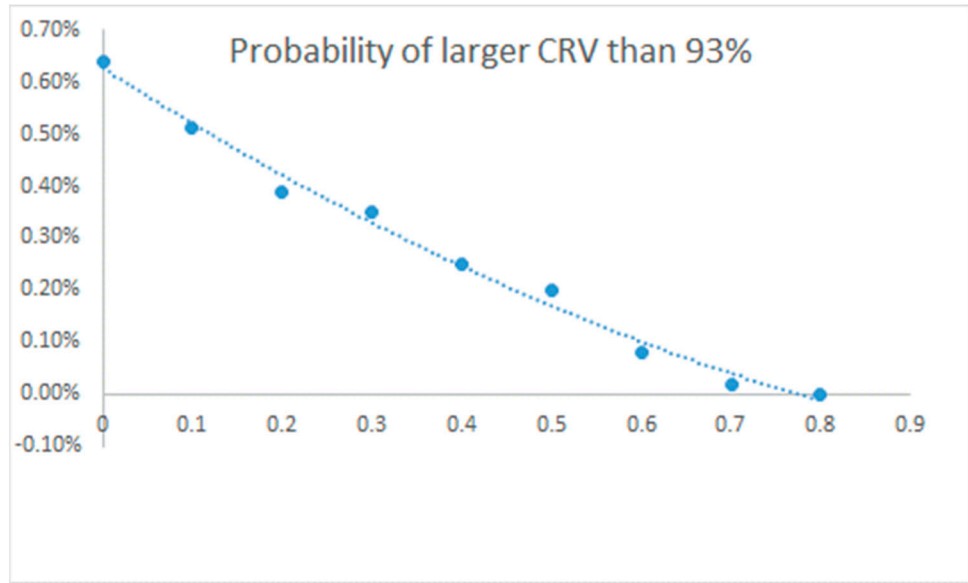

**Figure 3.** Class variance ratio.

Figure 3 shows the likelihood of CVR exceeding the threshold of 93%, depending on the marginal bias. Although marginal, it becomes rarer with our sample size as the bias increases. The minor non-smooth behavior of the curve is due to its compilation by means of simulation contrary to a closed-form formula.

The result of this analysis supports the validity of our esg2go framework, despite the fact that subsequent data will improve the framework considerably. Some residual bias implies the necessity of more model granularity for the sake of comparability and fair scoring. There might also be unexpected events which deliver unlucky or lucky scores, respectively. Although this kind of model risk is captured to a certain extent by our measures, for example, by the error tolerance intervals of our scoring function, we think that increasing model

granularity must be an ongoing task, which must be continuously examined as more data becomes available and by applying knowledge and methods from actuarial sciences, combined with machine learning techniques.

## 6. Discussion

Stricter sustainability and climate reporting requirements, especially in Europe and the United States, aim at increasing accountability and transparency in relation to a firm's impact on society and the environment in general and climate change in particular. However, the established global providers of ESG ratings have come under increased scrutiny in recent years since their main source of revenue tends to be consulting, which represents a conflict of interest, especially when they are also involved in verification [71]. In addition, an increasing skepticism is spreading about what ESG ratings actually measure, since the scores primarily assess how well a company manages its ESG risks to its own bottom line. This type of approach does not reveal how the actual sustainability performance of a company compares to its peers [54,62].

Recent empirical studies have revealed several rating biases [34–36,39–41] and a great variation in rating outcomes depending on the choice of rating and its underlying definition of corporate sustainability [18,19,21,23–25]. Other studies point out that a risk-based approach used in ESG rating and carbon footprint measurement may lead to a withdrawal of foreign direct investment from high-risk low-income countries and thus lead to exclusive growth rather than the inclusive growth that the UN Sustainable Development Goals (UN SDGs) envision [29,59]. Finally, several studies have revealed that current ESG rating and reporting systems are time-consuming and costly, and therefore not suitable for SMEs [7,53,65].

esg2go started as small project in 2018 with the primary purpose of developing a practical ESG rating prototype tool for Swiss-based SMEs. The practicality was ensured by working with the companies that use the tool through numerous feedback loops. In this context, many companies pointed out that the initial rating did not reveal anything about the positive impact they may generate through their solution-oriented businesses. We therefore added an optional input that gives SMEs the opportunity to highlight such positive impacts of their core business activity in the form of a handprint assessment [31,56].

In the second stage, esg2go was developed into a user-friendly rating tool designed to address the deficiencies of existing ESG evaluations through a calibrated benchmarking system that is based on a commonly agreed on and pragmatic definition of corporate sustainability as the 'ability of a company to coexist'.

A preliminary analysis of the data entered so far has revealed that the scoring function is able to minimize variation in SMEs to a level that makes the measurability and comparability of sustainability performance across sectors and firm size possible. As such, esg2go provides firms with a practical and credible sustainability certificate that can be made compatible with any other rating and reporting standard requested by clients or regulators thanks to the use of a filter technology. Moreover, esg2go also serves as an internal dashboard for companies to track their sustainability performance over time.

esg2go is the first rating tool that allows a company to compare its sustainability performance with its peers. The semi-automated esg2go rating report then enables a company to analyze the rating outcome in each key area and to define measures and set performance targets to improve its scores. This allows for a continuous, transparent, and fact-based tracking of sustainability performance in different key areas over time. It enables a science-based and differentiated long-term discussion with stakeholders about sustainability performance. Finally, esg2go does not face any conflict of interest since it consists of an automated rating only. Consulting is performed by independent certified professionals and verification is possible by any audit firm as long as it applies the verification concept developed by SQS and SGES. As such, esg2go shifts attention from identified problems in ESG rating and reporting to concrete solutions.

### 7. Concluding Remarks

The esg2go framework builds on two basic insights: a sustainability rating framework must be sustainable by itself, and sustainability management has to be understood as a form of risk management. In this paper, we employed these insights in the development of the esg2go rating methodology. It is based on an adaptive parametrization in which companies are assigned to specific sustainability classes formed by criteria such as the firm size and service category.

The weighing process of the KPIs is derived from a coherent and pragmatic definition of SME sustainability as the 'ability to coexist'—the extent to which a firm is able to meet the expectations of their most important stakeholders. These expectations have changed in recent years in view of a new and stricter regulatory environment concerning sustainability reporting and ESG due diligence procedures. For now, the regulation affects primarily larger companies, but they also indirectly affect SMEs as they are increasingly asked by their important clients to present a sustainability certificate. esg2go is a practical tool that allows firms to obtain a first assessment of their corporate sustainability performance, based on which they can track their improvements in different key areas over time or compare their performance to comparable others in their class. A preliminary empirical analysis in this paper has shown that the rating tool is able to minimize bias and produce a fair and robust assessment. The framework is not yet perfect. Due to its vision to be a learning instrument for the user and developer, it chases the optimal state by means of available information and will improve over time as more data from SMEs become available.

**Supplementary Materials:** The following supporting information can be downloaded at: https://www.mdpi.com/article/10.3390/su152416872/s1, File S1: Supplementary Material esg2go.

**Author Contributions:** The article was conceptualized and drafted by P.A. and I.C. M.M.B. reviewed and revised the article and supported the original draft preparation. The methodology and formal analysis were developed and carried out by I.C. and B.C. All authors have read and agreed to the published version of the manuscript.

**Funding:** The base funding of CCRS enabled method development and the resulting publication.

**Institutional Review Board Statement:** Not applicable.

**Informed Consent Statement:** Not applicable.

**Data Availability Statement:** The data presented in this study are available on request from the corresponding author.

**Acknowledgments:** We would like to express our gratefulness to Luzi Rageth, CEO of Adjumed Services AG, who converted our rating methodology into a user-friendly online product. As a leading provider of registry technology, Adjumed Services has been committed to transparency and quality in medical quality assurance in the health care sector for over 20 years.

**Conflicts of Interest:** The author declares no conflict of interest.

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
