# Peer review of "esg2go: A Method to Reduce Bias, Improve Coherence, and Increase Practicality of ESG Rating and Reporting"

_sustainability, doi:10.3390/su152416872_

Round 1

Reviewer 1 Report

Comments and Suggestions for Authors

The article addresses the very important and very current issue of ESG rating and reporting. It is because of these two characteristics and the regulations derived from the Green Deal that there is a huge need for a methodology for assessing and reporting progress in the field of ESG. As a result, a huge market for services in this area is created, because especially SMEs are unable to cope with these tasks on their own. This, in turn, creates a risk of bias and lack of consistency in the ESG rating and reporting process. The ambition of the authors of the reviewed article was to demonstrate that the esg2go method they presented at least reduces these threats. The structure of this article is focused on  this goal. The article is largely intended to promote this method. It arouses the reader's interest, but its disadvantage is that it shows too little the details of this method. Some theses, often quoted after other publications, are polemical. For example, that it is stakeholder expectations that may reflect social rather than economic demands. The essence of sustainable development is maintaining a balance between environment, society and economics. The authors themselves recognize this, saying that esg2go has the win-win potential of sustainability. The reader is also dissatisfied with the lack of specific, even exemplary, indicators that esg2go takes into account. The same applies to the level of KPI values, based on which it is assessed whether these values are satisfactory. Although I believe that the article can be published in its current form, I suggest the authors remove these weaknesses of the article for the sake of the reader.

Author Response

We would like to thank the reviewer for the very valuable comments to the earlier draft of our manuscript. We revised it accordingly and would like to respond in detail to the three main concerns raised by the reviewer (see document attached)

Reviewer 2 Report

Comments and Suggestions for Authors

Literature review on the topic is adequate. There is good background introduction provided with developments in the area of ESG reporting well analysed and referenced. There is also good justification for the paper. 

While information provided in Section 3 and 4 that explain on the scoring and rationale of esg2go system are relevant and adequate, the structure and presentation sequence may require major re-structuring. The current flow of presentation is quite difficult to understand and follow. It is difficult for a reader to understand section 3 before knowing the scoring items and details presented in section 4.

Another example of inadequate information flow is the use of "handprint". It is not clearly explained in section 1 when it is first used but was explained later in section 4. 

It is also suggested that shorter and clearer headings and sub-headings be used to provide clearer guidance for reader.

Comments on the Quality of English Language

Generally, well-written. Minor typo error - example: 1) line 199 "farm size" - firm size. 2) line 230 and 231 - the use of "it" needs clarification to avoid ambiguity 

Author Response

We would like to thank the reviewer for the valuable comments, which greatly helped to further improve the manuscript. A detailed response to the points raised by the reviewer and how we addressed them is attached.

Round 2

Reviewer 2 Report

Comments and Suggestions for Authors

The revised version is much clearer and has addressed the previously identified concerns. Thanks for the great efforts.